# Surface Enhancement Using Black Coatings for Sensor Applications

**DOI:** 10.3390/nano12234297

**Published:** 2022-12-03

**Authors:** Martin Hruška, Joris More-Chevalier, Přemysl Fitl, Michal Novotný, Petr Hruška, Dejan Prokop, Petr Pokorný, Jan Kejzlar, Virginie Gadenne, Lionel Patrone, Martin Vrňata, Jan Lančok

**Affiliations:** 1Department of Physics and Measurements, University of Chemistry and Technology Prague, Technicka 5, 166 28 Prague, Czech Republic; 2Institute of Physics, Czech Academy of Sciences, Na Slovance 2, 182 21 Prague, Czech Republic; 3Faculty of Mathematics and Physics, Charles University, V Holesovickach 2, 180 00 Prague, Czech Republic; 4ISEN Yncréa Méditerranée, Aix Marseille Univ, Université de Toulon, CNRS, IM2NP, 83000 Toulon, France

**Keywords:** nanostructured materials, black aluminium, black gold, QCM sensors, sensor applications, sputtering depositions, evaporation depositions

## Abstract

The resolution of a quartz crystal microbalance (QCM) is particularly crucial for gas sensor applications where low concentrations are detected. This resolution can be improved by increasing the effective surface of QCM electrodes and, thereby, enhancing their sensitivity. For this purpose, various researchers have investigated the use of micro-structured materials with promising results. Herein, we propose the use of easy-to-manufacture metal blacks that are highly structured even on a nanoscale level and thus provide more bonding sites for gas analytes. Two different black metals with thicknesses of 280 nm, black aluminum (B-Al) and black gold (B-Au), were deposited onto the sensor surface to improve the sensitivity following the Sauerbrey equation. Both layers present a high surface roughness due to their cauliflower morphology structure. A high response (i.e., resonant frequency shift) of these QCM sensors coated with a black metal layer was obtained. Two gaseous analytes, H_2_O vapor and EtOH vapor, at different concentrations, are tested, and a distinct improvement of sensitivity is observed for the QCM sensors coated with a black metal layer compared to the blank ones, without strong side effects on resonance frequency stability or mechanical quality factor. An approximately 10 times higher sensitivity to EtOH gas is reported for the QCM coated with a black gold layer compared to the blank QCM sensor.

## 1. Introduction

Quartz crystal microbalances (QCM) sensors refer to a quartz resonator that oscillates at a characteristic resonant frequency. It stands out as a direct, label-free detection tool suitable for real-time monitoring. Frequency shifts are induced by the changes in resonator mass as a result of the surface adsorption of molecules. The QCM sensors are based on the piezoelectric properties of quartz. Mechanical deflections are produced by applying an alternating electrical potential across the quartz-crystal cut. The frequency drop $\Delta f_{m}$ caused by an applied mass $\Delta m_{s}$ is described by the Sauerbrey equation [1]. If the active surface of the sensor is increased, then the applied mass also has the potential to increase, which, consequently, can lead to a higher value of $\Delta f_{m}$, thus improving the sensitivity. To enhance the selectivity, it is possible to decorate the active layer of the QCM with certain specific receptors, e.g., antibodies.

A sensitive biosensor can be obtained with a combination of specialized antibodies—from several review journals [2,3,4]. QCM sensors have been used for the detection of volatile organic compounds (VOCs), which are one of the causes of air pollution problems [5,6,7,8]. Sensitive and selective polymer films deposited onto the QCM electrodes have been used as an adsorption surface for these VOC vapors. Several applications have been reported in both gas and liquid phases for odorant biosensors [9,10]. Recently, an improvement in the discrimination of QCM gas sensors has been reported using coated sensing polymeric films such as ethyl cellulose, polymethyl methacrylate, Apiezon L, and T [6]. Moreover, 5.73 times higher responses have been shown for the 30-MHz sensors than for the 12-MHz ones.

Black metals (BMs) are defined as highly porous nanocrystalline materials that can significantly trap incident light [11,12,13]. These porosities are introduced during the growth of the metal film due to the presence of impurities which creates a structure allowing complex subwavelength electromagnetic interactions with the light in wide wavebands [14]. To date, BMs have found applications in electronics for optical sensing and imaging [14,15,16], heat radiators enhancement [17,18,19], electrochemical sensing and catalysis [20], solar cells [21], and energy harvesting [11,14,22,23,24,25]. Especially highly porous aluminum has attracted great interest in recent years [26,27]. Several metals, such as gold, platinum, tungsten, copper, titanium, palladium, or aluminum, were examined to obtain a black or colored coating [13,28,29,30,31,32,33,34,35]. Recently, surface desorption processes have been reported on thin layers of black aluminum [35]. Desorption measurements reported that weakly bound atoms of oxygen and nitrogen have two stages of desorption: one below room temperature (RT) corresponding to physical desorption and the latter above RT corresponding to desorption of chemisorbed particles.

In this work, we report the sensitivity improvement of QCM sensors coated by BMs layers. Two different BMs are deposited on QCM sensors, black aluminum (B-Al), and black gold (B-Au). The consequences between their high roughness and the improvement of QCM sensor characteristics (sensitivity, quality factor, equivalent circuit parameters) are discussed.

## 2. Experimental

### 2.1. QCM Sensor Substrates

The specially prepared QCMs resonators with gold electrodes from *Krystaly Hradec Kralove a.s.* were used as QCM sensor substrates. These QCMs are AT-cut ones with a resonator diameter of 8.65 mm and electrode diameter of 4.4 mm, oscillating at a fundamental frequency of 10.88 MHz. Their operating temperature range is between −20 °C and +70 °C, with a quality factor (Q) of ≥45,000. For black metal preparation, a circular-shaped homemade mask was used.

### 2.2. Preparation of Black Aluminium

The black aluminum (B-Al) films were deposited by pulsed DC magnetron sputtering. A DC power supply *Hüttinger 3000* combined with a pulse generator *MELEC* was operated at the power of 400 W. The repetition rate was set to 10 kHz with a duty cycle of 0.5. An Aluminium target (99.99% purity) with a diameter of 100 mm was used for sputtering. A distance of 100 mm was fixed between the target and the substrate. The base pressure in the chamber of $5\times 10^{-3}$ Pa was ensured by the diffusion pump, and the magnetron discharge was maintained in N_2_/Ar mixture atmosphere at a constant total pressure of 0.5 Pa, which was regulated by a throttle valve situated at the high-vacuum pump gate. Ar flow was fixed at 16 sccm, and the N_2_ flow was kept at ~0.4 scc, which corresponds to N_2_/Ar mixture of ~6.5%. The thicknesses of the deposited films were 280 nm.

### 2.3. Preparation of Black Gold

The black gold (B-Au) films were prepared by thermal evaporation method from a tungsten boat in an inert argon atmosphere using *Zahoxin KXN-15200D DC power supply.* The samples were mounted on a stainless-steel table at a distance of 60 mm above the heating source. A pure 100 mg gold pellet was the source material for evaporation. The deposition was performed in five steps with a heating current of 130 A; it was regulated so that the temperature of the substrates did not exceed 50 °C. It was experimentally observed that this is the maximal possible temperature that results in a pitchy black-gold layer. The deposition conditions are summarized in Table 1.

### 2.4. Instrumentation and Devices

#### 2.4.1. Scanning Electron Microscopy

Surface morphology was characterized by scanning electron microscopy (SEM) using *Mira 3 Tescan electron microscope* (TESCAN Inc., Brno, Czech Republic) at high magnifications (100 kx and 500 kx) with a perpendicular in-beam secondary electron detector at an accelerating voltage of 10 kV and 30 kV, respectively. The layer thickness was determined by sample cross-section measurement starting from the cut of glass substrates.

#### 2.4.2. Atomic Force Microscopy

Atomic force microscopy *AFM Dimension ICON, Bruker,* and *Bruker Multimode 8* equipped with *Nanoscope V* electronics (Bruker Inc., Camarillo, CA, USA) was used to investigate the surface morphology and roughness. Measurements were made under ambient conditions, and images were obtained by Peak Force Tapping mode using *ScanAsystAir* tips with scan areas of 1 × 1 μm^2^.

#### 2.4.3. Impedance Spectroscopy of QCMs

Blank and deposited QCM sensors with black gold (B-Al) and black aluminum (B-Al) films were characterized by an *Agilent 4294A precision impedance analyser*, 40 Hz–110 MHz (Keysight Technologies, Inc., Santa Rosa, CA, USA). Impedance spectra were measured around the fundamental resonance frequency of the QCMs at room temperature and in a constant gas flow of $70\;{\mathrm{mL}\;\min}^{-1}$ of synthetic air. A homemade 4-wire probe adapter and glass chamber were used for the measurement. From the obtained spectra, the parameters of the equivalent circuit, resonance frequencies, and quality factor were determined using an internal fitting algorithm of the impedance analyzer.

#### 2.4.4. Measurement of Sensor Response

The characteristics of the QCM sensor were measured in a homemade ground glass joint apparatus that is capable of simultaneous measurement of up to 4 QCMs in a cascade-like arrangement. The principal configuration of the measuring setup is shown in Figure 1. The dead volume of the glass chamber is approximately 2.5 mL. To determine the resonance frequency of QCMs, a home-developed oscillator circuit with inverter chips driven at 5 V d.c. in combination with *National Instrument PCI-6602*,8-channel counter card (National Instruments Inc., Austin, Texas, USA) was used. The QCM sensors were measured at room temperature, in a constant gas flow rate of $70\;{\mathrm{mL}\;\min}^{-1}$ using tedlar bags for analytes and reference atmosphere. All sensors were measured simultaneously, together with a blank QCM as a reference. The temperature was measured with a *PT100* probe. Synthetic air was used as the reference atmosphere; the measured mixture contained a defined concentration of ethanol or water vapors, respectively.

## 3. Results and Discussion

### 3.1. Investigation of Morphology and Thickness of Black Metal Films by SEM

SEM images from the B-Al layer and B-Au layer deposited on the surfaces of the QCM sensor are presented in Figure 2, which shows the surface of the B-Al and B-Au layers; Figure 3 presents cross-section images of both layers. Both surface morphologies reveal similar structures of cauliflowers similar to those reported in the literature [34,36,37]. The two main differences are the size of the grains and the layer densities, which are smaller for B-Au than the B-Al. The cross-section pictures in Figure 3 illustrate these differences, where the B-Al film presents pretty dense columnar structures, while the B-Au film presents a highly porous columnar structure. These differences originate from the different deposition techniques used to coat the QCM sensors. The energy of particles from the DC pulsed sputtering technique is higher than the energy from the evaporation technique, which leads to a higher energetical film growth process in the first case. Both techniques have been used to deposit black metal films, and similar results have been reported to those observed in this work [36,37]. The B-Au films deposited by evaporation presented highly porous surfaces made of condensed particles with a diameter between 5 and 20 nm [37]. A chain-like structure has been described due to the high porosity that also corresponds to the B-Au film morphology reported in this work and observed in Figure 2. The B-Au samples prepared by sputtering techniques showed less porous surface films made of condensed particles with a diameter of ~60 nm [36]. The prepared black metal layers, according to the morphology investigated, match the black layers prepared previously; therefore, a more detailed characterization of the physical properties can be found in the previous articles [12,31]. The thickness of the black aluminum layer and the black gold layer was estimated to be 278 ± 11 nm and 280 ± 40 nm, respectively.

### 3.2. Atomic Force Microscopy

AFM pictures are shown in Figure 4. The B-Al surface is presented in Figure 4a, and the B-Au surface is shown in Figure 4b. RMS values of 49 nm and 305 nm are measured from the surface roughnesses of B-Al and B-Au films, respectively. The RMS value for B-Al is lower than those reported in our previous works, which were equal to 93 and 114 nm, respectively [11,12]. The mean difference between each result is the thickness of B-Al layers deposited previously of 500 nm and 1 µm, which are higher than the one deposited on the QCM sensors of 280 nm. An increase in the roughness value can be correlated with an increase in the B-Al film thickness.

A six-times higher RMS value is measured from the B-Au surface than from the B-Al surface, which confirms the higher roughness of the B-Au layer deposited on the QCM sensor. The presence of agglomerated grains of ~20 nm diameter can be observed on the surface of the B-Au film shown in Figure 4b. It also confirms the chain-like structure already observed in the SEM images in Figure 3a,b, which leads to a cauliflower surface morphology of the film.

### 3.3. Impedance Spectra and Stability of QCM Sensors

The impedance spectra of the QCM sensors before and after the deposition of the B-Al layer and the B-Au layer are presented in Figure 5a,b. Parameters of the commonly used Butterworth–Van Dyke (BVD) equivalent circuit, modeled by the impedance analyzer, were used for the QCM characterization. They are presented in Table 2 and Table 3. Since we are operating QCMs under atmospheric conditions, the BVD model is sufficient for the characterization of prepared QCM sensors, and its parameters well describe the behavior of the resonator. According to the literature, the motional resistance $\left(R_{s}\right)$ of equivalent circuit matches the dissipation of the oscillation energy with the influence of the surrounding environment, which is in contact with the crystal. The capacity $\left(C_{s}\right)$ corresponds to the energy stored in the oscillation and is related to the elasticity of quartz and surrounding spaces, while the inductance $\left(L_{s}\right)$, in turn, corresponds to the mass loading adsorbed on the surface during the vibrations. The capacity $\left(C_{0}\right)$ is determined primarily by the capacitor, formed by the gold electrodes on both sides of a QCM, and depends mainly on its geometry [38].

It is clear that to ensure the correct operation of the QCM sensor, the stability of oscillations plays a significant role. In terms of oscillations stability, the motional resistance $\left(R_{s}\right)$ and the quality factor (Q), which is determined by Equation (1), are commonly used as the main parameters to assess the stability of QCM resonators regarding the load of the deposited layer [39,40]. Suppose the quality factor is too low or the motional resistance is too high. In that case, dumping occurs, so it becomes impossible to determine the resonance frequency, especially by the common oscillating drive circuits.
(1)$$Q=\frac{1}{2\pi \cdot f_{s}\cdot R_{s}\cdot C_{s}}$$

Regarding these circumstances, the layers of the black coatings were kept relatively thin (280 nm) so that the quality factors (Q) are still relatively high ($Q_{\mathrm{BAl}}=1.7\cdot 10^{4}$ for black aluminum and $Q_{\mathrm{BAu}}=3.4\cdot 10^{4}$ for black gold) in comparison to the quality factors of blank sensors ($Q_{\mathrm{Blank}}=5.8\cdot 10^{4}$ and $5.2\cdot 10^{4}$) The oscillations are, therefore, stable, which is also supported by the motional resistance $\left(R_{s}\right)$, which is commonly around 10 Ω.

The change in resonance frequency of the QCM ($\Delta f_{s}$) was used to determine the total deposited mass and the density of the deposited black metal layers, according to [41]. The following equation was used:(2)$$\Delta m=\frac{A\cdot \sqrt{\rho_{q}\cdot \mu_{q}}}{2f_{0}{}^{2}}\Delta f_{s}$$
where ($\Delta f_{s}$) is the measured frequency shift due to the deposited mass, ($f_{0}$) is the fundamental series resonant frequency of the blank QCM ($f_{s}$), (A) is the electrode area and ($\rho_{q}$) and ($\mu_{q}$) are the density and shear modulus of quartz, respectively. The deposited mass of the black gold layer (B-Au) was estimated to 12.5 μg and its density to $2.9\;{g\;\mathrm{cm}}^{-3}$, which is almost 10 times lower than the density of bulk gold. This is in correlation with the highly porous surface measured by SEM and AFM. In the literature, one can find even lower densities, but that is the case for thicker layers—up to tens of microns, e.g., [37]. The deposited mass of the black aluminum (B-Al) layer was estimated to 11.8 μg and the density to $2.7\;{g\;\mathrm{cm}}^{-3}$, which is the same as for the bulk material.

### 3.4. Gas Sensor Measurements

To investigate the sensing properties of prepared black metal layers and to determine the effect of highly nanostructured surfaces, the frequency response of prepared QCM sensors was simultaneously measured with the blank QCM, serving as a reference, to different concentrations of EtOH and water vapors. Figure 6a depicts the response to water vapors in terms of different relative humidity, while Figure 6b depicts the response to different concentrations of EtOH vapors. It can be clearly seen that the black metal coatings increased the response of the QCM sensors compared to the reference blank in both cases.

To determine the effect of the nanostructured black metal coating, the standard calibration curve was used, and the sensitivity, computed as a slope of the calibration curve (according to IUPAC), served as the main parameter [42]. Calibration curves for relative humidity and ethanol vapors are depicted in Figure 7. The computed sensitivities (S) and also sensitivity factor (Sf), that was determined as the ratio of the black coated sensor sensitivity to the sensitivity of blank QCM(3), are listed in Table 4.
(3)$$Sf=\frac{S_{B-M}}{S_{\mathrm{Blank}}}$$

As can be seen, the black aluminum coating increased the sensitivity of QCM to water and ethanol vapors by 2.4 times and 2.6 times, respectively, while the black gold coating increased the sensitivity of QCM by a factor of 5.2 for water and 10.2 for EtOH. The increase in sensitivity of QCMs by porous gold films was already reported in the literature with different results depending on the used analytes [43,44]. Enhanced sensitivities of 1.4 for N_2_, He, and 2 for SF_6_ [42] were reported for QCM sensors coated with Au-Ag porous electrodes operating at 300 K. This enhancement was even able to be a factor of 40 for liquid N_2_, He detection. Porous Au deposited by electrochemical technique was also used to improve the sensitivity of QCM for biosensing [43]. Here, the authors achieved improvement by a factor of 3 in response to myoglobin.

Regarding the interaction of black metals with gas species, it is clear that the nanostructured surface of black gold is highly stable; on the other hand, the porous surface of black aluminum tends to oxidize spontaneously, especially in the surface layer [45]. However, this oxidation takes place mainly after the preparation of B-Al, when the samples are removed from the chamber into the atmosphere. As depicted in Figure 6, even the QCM with the B-Al layer showed good long-term stability during the measurement that was carried out several days after the sensor preparation. The stability of the B-Al surface is also confirmed by previous thermally stimulated desorption measurements [35] and XPS measurements of the nanostructured surface [11].

In this study, we focus mainly on the comparison of black coatings with different porosities. It is clear that highly nanostructured coatings—such as black metal—can significantly increase the sensitivity of QCM sensors, and this effect is highly affected by the surface morphology of black metal coatings. The black gold layer shows a more porous surface than the black aluminum one and, therefore, provides more bonding sites for the gas analytes and has the potential for a larger increase in sensitivity. As for the selectivity of obtained sensors—although the black metal layers themselves are not very selective, there is still big potential to use them as sensitivity enhancers. The QCM electrode covered by black metal can be subsequently functionalized by proper selective receptors that decorate the nanostructured surface of black metal and provide better selectivity while still preserving increased sensitivity. Thus, e.g., porous and nano-porous gold electrodes were functionalized using self-assembled monolayers (SAMs) by protein arrays for the detection of peptides in blood plasma [46] or used as biosensors for liquid detection of organophosphates [47]. We have also published the first attempts at the functionalization of black gold layers by SAMs with promising results [48].

## 4. Conclusions

Black metal coatings were utilized to improve the sensitivity of the QCM sensor for gas detection. Two metals have been tested, black gold (B-Au) and black aluminum (B-Al), with a film thickness of 280 nm for both. Each film showed a highly porous structure with a large relative surface made up of condensed particles. These porosities are introduced during the film metal growth due to the presence of impurities. The sensitivity of sensors has been tested on H_2_O vapors and EtOH vapors at different concentrations. An improvement of QCM sensor factors of 5.3 and 4.1 for B-Al coated QCM and of 21.3 and 8.8 for B-Au coated QCM was observed under H_2_O and EtOH vapors, respectively. A clear improvement has been reported in comparison to the blank QCM sensor, showing the advantage of coating QCM with black metals for gas sensing applications.

## Figures and Tables

**Figure 1 nanomaterials-12-04297-f001:**
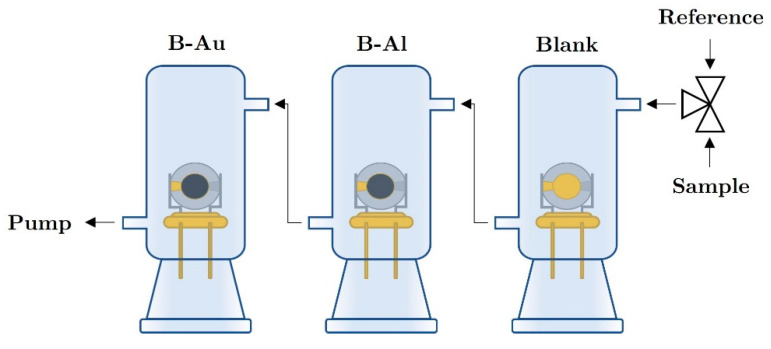
Scheme of the measurement apparatus.

**Figure 2 nanomaterials-12-04297-f002:**
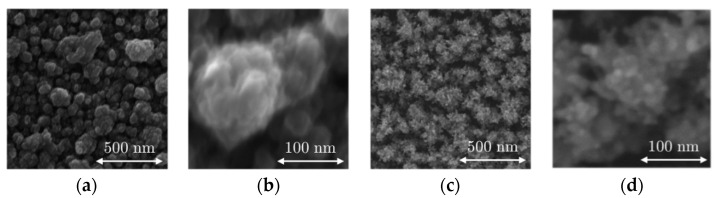
SEM images of black aluminum (B-Al) (**a**) 100 kx, (**b**) 500 kx, and black gold (B-Au) (**c**) 100 kx, (**d**) 500 kx.

**Figure 3 nanomaterials-12-04297-f003:**
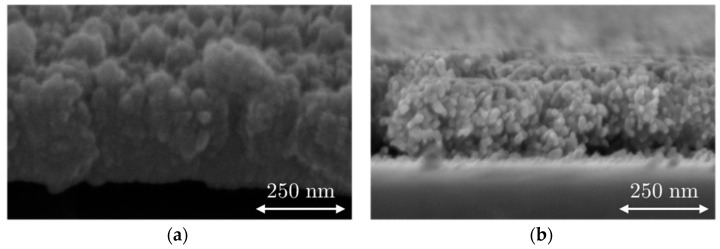
SEM images of the B-Al layer (**a**,**c**) and the B-Au layer (**b**,**d**) deposited on QCM sensors. (**c**) Cross section of B-Al layer and (**b**) cross-section of the B-Au layer.

**Figure 4 nanomaterials-12-04297-f004:**
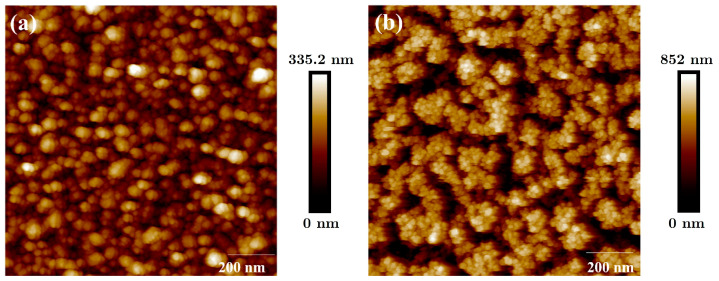
AFM pictures of B-Al surface layer in (**a**) and B-Au surface layer in (**b**) deposited on QCM sensors.

**Figure 5 nanomaterials-12-04297-f005:**
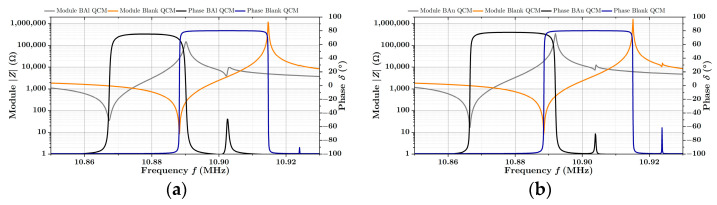
Impedance spectrum of QCM sensors before and after coating with B-Al layer in (**a**) and before and after coating with B-Au layer (**b**).

**Figure 6 nanomaterials-12-04297-f006:**
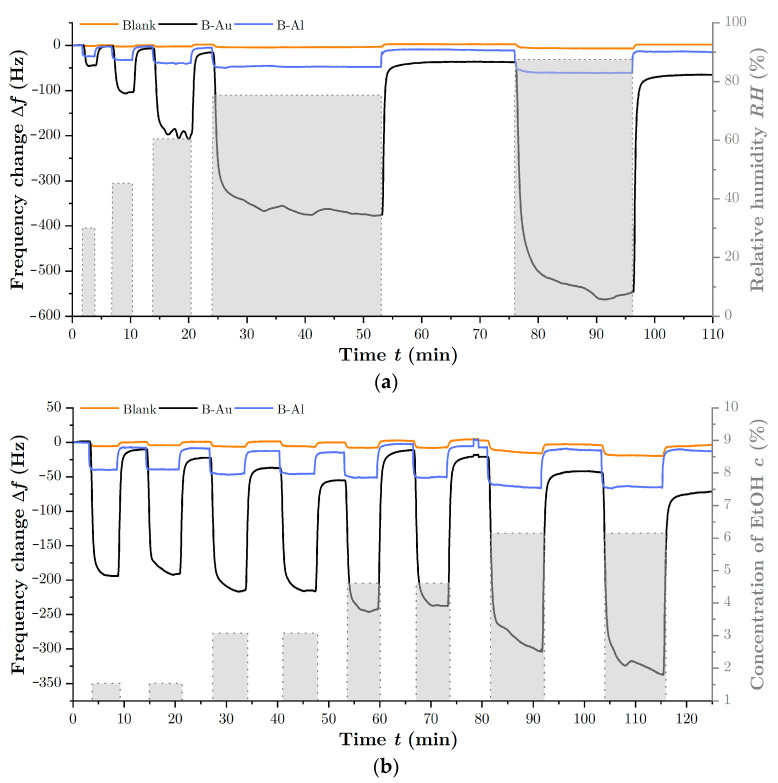
Response of QCM sensors with and without BMs coatings as a function of the concentration of H_2_O vapor in (**a**) and as a function of the concentration of EtOH vapor in (**b**).

**Figure 7 nanomaterials-12-04297-f007:**
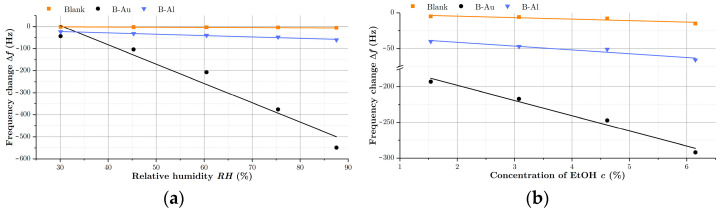
Calibration curves and sensor sensitivity factors from sensor responses to H_2_O in (**a**) and towards vapors of EtOH in (**b**).

**Table 1 nanomaterials-12-04297-t001:** Deposition conditions of B-Au.

Quantity	Value
Base Pressure	$5.5\cdot 10^{-4}\;\mathrm{Pa}$
Working Pressure	100 Pa
Heat Power	<340 W
Substrate Temperature	<50 °C

**Table 2 nanomaterials-12-04297-t002:** Equivalent circuit parameters for QCM with B-Al layer.

$R_{s}\;\left(\Omega \right)$	$C_{s}\;\left(\mathrm{fF}\right)$	$L_{s}\;\left(\mathrm{mH}\right)$	$C_{0}\;\left(\mathrm{pF}\right)$	$f_{s}\;\left(\mathrm{Hz}\right)$	$f_{p}\;\left(\mathrm{Hz}\right)$	Q (1)
10.4	24.2	8.8	5.0	10,888,252.49	10,914,695.32	$5.8\cdot 10^{4}$
35.4	24.3	8.8	5.8	10,867,364.74	10,890,115.17	$1.7\cdot 10^{4}$
			$\Delta f_{s}$	20,887.74		

**Table 3 nanomaterials-12-04297-t003:** Equivalent circuit parameters for QCM with B-Au layer.

$R_{s}\;\left(\Omega \right)$	$C_{s}\;\left(\mathrm{fF}\right)$	$L_{s}\;\left(\mathrm{mH}\right)$	$C_{0}\;\left(\mathrm{pF}\right)$	$f_{s}\;\left(\mathrm{Hz}\right)$	$f_{p}\;\left(\mathrm{Hz}\right)$	Q (1)
10.6	26.3	8.1	5.4	10,888,549.48	10,915,112.63	$5.2\cdot 10^{4}$
18.0	24.2	8.9	5.2	10,866,520.86	10,891,966.94	$3.4\cdot 10^{4}$
			$\Delta f_{s}$	22,028.61		

**Table 4 nanomaterials-12-04297-t004:** Sensitivity of prepared QCMs with black metal coatings.

	Ethanol Vapours		Water Vapours	
QCM Sensor	Sensitivity $\left(S_{\mathrm{EtOH}}\right)$	Sensitivity Factor $\left(Sf_{\mathrm{EtOH}}\right)$	Sensitivity $\left(S_{H_{2}O}\right)$	Sensitivity Factor $\left(Sf_{H_{2}O}\right)$
Blank	2.1	1.0	1.7	1.0
B-Au	21.3	10.2	8.8	5.2
B-Al	5.3	2.6	4.1	2.4

## Data Availability

In this section, please provide details regarding where data supporting reported results can be found, including links to publicly archived datasets analyzed or generated during the study.

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
