# Peer review of "Surface Enhancement Using Black Coatings for Sensor Applications"

_nanomaterials, 2022, doi:10.3390/nano12234297_

Round 1
Reviewer 1 Report
The manuscript entitled "Surface Enhancement Using Black Coatings for Sensor Applications" has been carefully examined. The authors' experimental results are clear. The microstructure of carbon black has a high surface structure, and the physical adsorption interaction strength of different gas molecules on structured carbon black is different. However, the research goals are not attractive enough for sensor applications. How to distinguish between water vapor and ethanol molecules? There are too few results presented in the manuscript, and there is also a lack of further exploration of the results. Therefore, I think that the paper in current format should not be published.
Author Response
We hope that we have clearly articulated that we are using highly nanostructured and nanoporous surfaces of metals (aluminium and gold) in the submitted draft. These coatings are also called black metals due to their black appearance.
The first reviewer made a good point about the selectivity of black metal coatings, he is right that these nanostructured layers themselves do not provide sufficient selectivity for main sensor applications. Nevertheless, herein we propose to use black metals just as sensitivity enhancers. Black metals can also be subsequently functionalized by other selective receptors for example by self-assembled monolayers, and we have already done so, but this is not the aim of this article. [1-3] We believe that this is just a part of a broader study about sensor use of black metals that is anyhow still very interesting for the audience of the Nanomaterial journal. As a response, we have included a wider discussion with references to the possibilities of functionalization of black metals (lines 263-276, text in blue color).
- M. Evans-Nguyen, S. C. Tao, H. Zhu, and R. J. Cotter, “Protein arrays on patterned porous gold substrates interrogated with mass spectrometry: Detection of peptides in plasma,” Anal. Chem., vol. 80, no. 5, pp. 1448–1458, Mar. 2008, doi: 10.1021/AC701800H.
- A. Hondred, Z. T. Johnson, and J. C. Claussen, “Nanoporous gold peel-and-stick biosensors created with etching inkjet maskless lithography for electrochemical pesticide monitoring with microfluidics,” J. Mater. Chem. C, vol. 8, no. 33, pp. 11376–11388, Aug. 2020, doi: 10.1039/D0TC01423K.
- Hruska, M.; Tomecek, D.; Havlova, S.; Fitl, P.; Guerkboukha, M., A.; Gadenne, V.; Patrone, L.; Vrnata, M. QCM Sensors Combining Highly Nanostructured Metal-Blacks Sublayers and Active Self-Assembled Monolayers, ECS Meet. Abstr., vol. MA2020-01, no. 31, pp. 2314–2314, May 2020, doi: 10.1149/MA2020-01312314mtgabs.
Reviewer 2 Report
In this paper black metals coatings were utilized to improve the sensitivity of QCM sensor for gas detection. Two metals have been tested, black gold (B-Au) and black aluminium (B-Al), with a film thickness of 280 nm for both. The authors claim a large highly porous surface made of condensed particles for both the coatings. These porosities are introduced during the film metal growth and the authors claim that this is due to the presence of impurities.
The sensitivity of sensors has been tested on H2O vapours and EtOH vapours at different concentrations. An improvement of QCM sensor was observed under H2O and ethanol vapours. The improvement has been reported in comparison to the blank QCM sensor showing the advantage to coat QCM with black metals for gas sensing applications.
The idea to use porous metal or porous metal oxide in gas sensing is not new at all and there are a large number of papers on the topic. the authors report here a reasonable number of references, but a better discussion on the state-of-the-art should be an improvement.
from the technical point of view the paper requires corrections in order to be considered for publication in Nanomaterials.
In particular:
line 233-240: the colors selected in the plot make really complicated to discriminate between Au and Al and in the text there are some chinese characters reporting an error.
techinally, using a porous metal and characterizing it only via SEM and AFM is very limiting. It should be much better to report a direct measure of the porosity and eventually a spectral measurement in order to demonstrate that is really black metal.
while porous (or highly rough as in this case) Au coating is easy to be prepared and is know to be stable, in the case of rough of porous Al the stability due to oxidation is a big issue and should be discussed. In particular the authors are using here B-Al with H2O and EtOH and both of them can oxide the film. porous Al (metallic) is an interesting metal very complicated to be obtained and a more detailed discussion could be a plus (in literature a couple of recent papers discussed on the preparation of porous metallic Al and should be mentioned here:
J. Electrochem. Soc. 2018, 165, C492–C496
Nanomaterials 2020, 10, 102
Author Response
As it was suggested by the reviewer, we have accordingly edited the plots (lines 235 - 236) and changed the colours of plotted data, therefore they are clearer now. We have also reviewed the text and it seems that Chinese characters were created after the conversion to PDF in the MDPI system, the meaning of Chinese characters relates to bad referencing in the manuscript. We have therefore inserted all the references as plain text. (These were just references to Figure 7 and Table 4).
Regarding the proper and more detailed characterization of black metal layers, these data have been already published by our colleagues in the literature. Therefore, we propose here just new data aiming mainly at the sensor application of black metals. From the SEM and AFM characterization is clear that the prepared layers match the previous ones that have been already characterized. To articulate this more clearly, we have added the relevant part to the discussion (lines 155 – 158, text in green color).
The reviewer is right that the porous aluminium films are much less stable than the gold ones due to the possible oxidation of the surface, that is actually the case. We, therefore, provided a more detailed discussion (lines 254-262, text in green color) and also expanded the introduction using suggested references.
We have also revised the English and its use by professional software.
Round 2
Reviewer 1 Report
Thanks to the author for his response to my comments. At present, the identification of the specificity of the sensor application is more important than the improvement of the sensitivity. Methyl ether, ethanol, and water all have excellent space-filling properties for the microstructure of metal black, but the sensor applications are limited to very narrow areas.
The authors' responses have been limited to sensitivity improvements, therefore, I have no further comments.
Reviewer 2 Report
The manuscript is now ok for publication